# The quantitative basis for the redistribution of immobile bacterial lipoproteins to division septa

Lara Connolley[1]☯, Joanna Szczepaniak[2]☯, Colin Kleanthous[2]*, Seán M. Murray[1]*

**1** Max Planck Institute for Terrestrial Microbiology and LOEWE Centre for Synthetic Microbiology (SYNMIKRO), Marburg, Germany, **2** Department of Biochemistry, University of Oxford, Oxford, United Kingdom

☯ These authors contributed equally to this work.
* colin.kleanthous@bioch.ox.ac.uk (CK); sean.murray@synmikro.mpi-marburg.mpg.de (SMM)

**Data Availability Statement:** The Matlab scripts for solving the models described in the text are available at https://github.com/lconnolley/Tol-Pal-model. The scripts for the SpatialFRAP technique

## Abstract

The spatial localisation of proteins is critical for most cellular function. In bacteria, this is typically achieved through capture by established landmark proteins. However, this requires that the protein is diffusive on the appropriate timescale. It is therefore unknown how the localisation of effectively immobile proteins is achieved. Here, we investigate the localisation to the division site of the slowly diffusing lipoprotein Pal, which anchors the outer membrane to the cell wall of Gram-negative bacteria. While the proton motive force-linked TolQRAB system is known to be required for this repositioning, the underlying mechanism is unresolved, especially given the very low mobility of Pal. We present a quantitative, mathematical model for Pal relocalisation in which dissociation of TolB-Pal complexes, powered by the proton motive force across the inner membrane, leads to the net transport of Pal along the outer membrane and its deposition at the division septum. We fit the model to experimental measurements of protein mobility and successfully test its predictions experimentally against mutant phenotypes. Our model not only explains a key aspect of cell division in Gram-negative bacteria, but also presents a physical mechanism for the transport of low-mobility proteins that may be applicable to multi-membrane organelles, such as mitochondria and chloroplasts.

## Author summary

In order for bacteria to successfully survive it is vital that they are able to concentrate proteins at precise sub-cellular locations. This usually occurs by capturing a freely diffusive protein, with the requirement that the protein is sufficiently mobile to reach the target location within the appropriate timeframe. Currently, it is unclear how immobile proteins are localised. Here, we examine how a very slowly diffusing protein Pal is able to localise to the centre of dividing cells to fulfil its role in membrane constriction. We present a mathematical model in which Pal is made mobile through the binding of another protein, TolB, and then deposited at the septum via active dissociation. This method is similar to a conveyor belt where Pal is collected everywhere but only deposited at the centre of the cell. We are able to fit this model to measurements of protein mobility and also test its

were described previously and are available at https://github.com/smury/SpatialFRAP.

**Funding:** CK acknowledges financial support for this work from the European Research Council (advanced grant 742555; OMPorg; https://ec.europa.eu/programmes/horizon2020/). SMM receives core support from the Max-Planck-Gesellschaft (https://www.mpg.de). The funders had no role in study design, data collection and analysis, decision to publish, or preparation of the manuscript.

**Competing interests:** The authors have declared that no competing interests exist.

predictions against mutant phenotypes. Our study is not only able to explain how this key protein is relocalised but also presents a general mechanism for the transport of slowly diffusing proteins that may be applicable to other systems.

## Introduction

The localisation of proteins to specific sites within the cell is critical for cellular function [1]. In bacteria, this localisation is typically achieved through passive mechanisms such as recruitment by some existing landmark (or inhibition by an anti-landmark), the sensing of membrane curvature or cell geometry [2]. The transport itself is typically purely diffusive and relies on the protein coming into spatial contact with its target factor ("diffusion and capture"). Thus the protein to be localised must be sufficiently mobile for its timely localisation to occur. However, membrane proteins are generally slowly diffusing [3], suggesting that active processes may be required for their relocalisation. Since the periplasm is not energised any active process would have to be powered by the proton motive force (PMF) across the inner membrane or ATP hydrolysis in the cytoplasm.

An example of such a protein is Pal (Fig 1A), a widely-conserved outer membrane lipoprotein found in Gram-negative bacteria that tethers the outer membrane to the cell wall via its peptidoglycan (PG) binding domain [6]. During cell division, constriction of the outer membrane must be coordinated with the invagination of the cell wall. This is thought to be achieved by recruitment of Pal to the division septum by the multi-protein Tol system, which localises to the constriction site as part of the division apparatus independently of Pal [7–9]. The Tol system consists of i) an inner membrane TolQ-TolR stator complex, ii) TolA, which is coupled to the TolQ-TolR stator, and iii) a periplasmic Pal-binding protein, TolB [7,9]. Binding of Pal to TolB prevents the former binding to PG. However, TolB can still interact with TolA even while in complex with Pal (Fig 1A). Based on molecular dynamic simulations [5], this relatively weak interaction is able to dissociate TolB from its complex with Pal by applying a pulling force. In recent work on *Escherichia coli*, we found that Pal diffuses extremely slowly with a diffusion coefficient measured by single particle tracking of $\leq 0.004$ μm$^2$s$^{-1}$ (likely an upper bound due to the limits of the technique) [5]. Such slow diffusion was attributed to the binding of Pal to the cell wall, rather than the embedding of its lipoylated domain in the inner leaflet of the outer membrane. Indeed lipoylated PAmCherry, displayed a diffusion coefficient more typical of membrane proteins of ~0.02 μm$^2$s$^{-1}$. This raised the question of how Pal is redistributed to the division septum if it diffuses so slowly.

To investigate this, we developed a new method, SpatialFRAP, for the analysis of Fluorescence Recovery After Photobleaching (FRAP) experiments [5]. It uses the Fokker-Planck equation for diffusion in homogeneous media, to assess the mobility of a protein across the length of the cell (Fig 1B and 1C). The output is $D_{eff}(x)$, a spatially-varying effective diffusion coefficient. We used this technique to measure Pal mobility in dividing (defined as having a visible constriction) and non-dividing (no visible constriction) cells. We discovered that, while still slow, Pal mobility in dividing cells is increased everywhere compared to non-dividing except at the septum, where it accumulates (Fig 1D). Thus, instead of Pal being made less mobile at the septum, which is the naively expected 'diffusion and capture' scenario, it is made more mobile everywhere else. Furthermore, we found this behaviour to be dependent on a functional Tol system. In particular, the aforementioned increase in mobility and resulting accumulation at the septum requires TolA, TolA coupling to the PMF and the binding of TolB to Pal and TolB to TolA [4,5]. To explain these results, we proposed that Pal mobility is the result of its binding to TolB in the periplasm (Fig 1A). Since this binding is mutually exclusive with

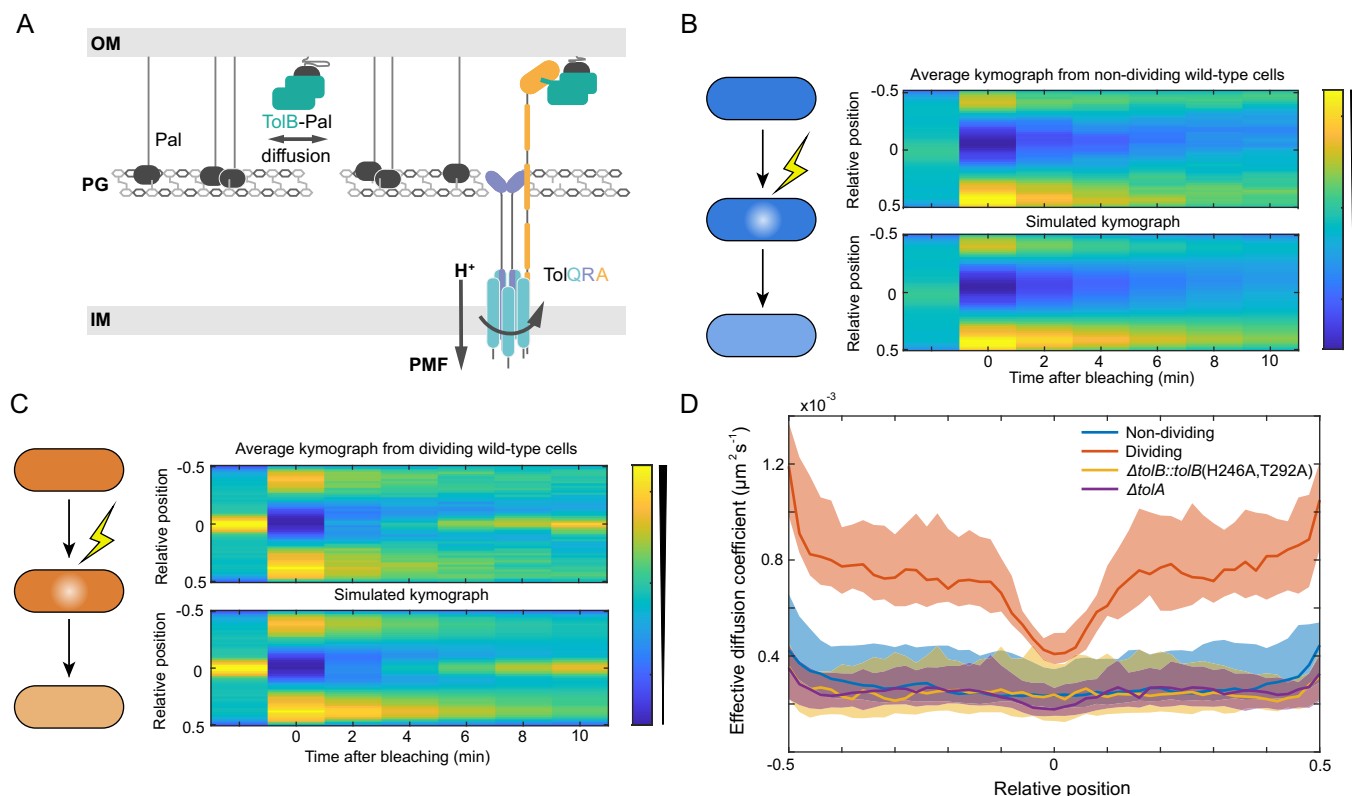

**Fig 1. Pal in dividing cells is mobilised by TolQRA.** (A) Major components of the Tol-Pal system. Pal is an outer membrane (OM) lipoprotein (black) that can bind *meso*-diaminopimelic acid within peptidoglycan (PG) or the periplasmic protein TolB (green) in a mutually exclusive manner. The inner membrane (IM) protein TolA spans the periplasm and is coupled to the PMF via its interaction with TolQ and TolR. TolA is also able to interact with TolB and can form a tripartite complex of TolA-TolB-Pal [4]. (B) Top, average kymograph of Fluorescence Recovery After Photobleaching (FRAP) of Pal-mCherry over 30 non-dividing wild-type cells where the colour scale indicates normalised fluorescence. Bottom, simulated kymograph obtained using the SpatialFRAP method by fitting to the data above. (C) Same as in (B) for 30 dividing wild-type cells. (D) The effective diffusion coefficient of Pal as a function of the relative cellular location in individual non-dividing, dividing, *ΔtolA*, and *ΔtolB::tolB*(H246A, T292A) cells. TolB H246A T292A is a mutant that is unable to bind Pal. Shown is the median as a function of relative long axis position with 95% confidence intervals for approximately 30 cells. Data in (B)-(D) reproduced from reference [5].

Pal binding to PG, TolB-Pal complexes would be expected to diffuse more rapidly. Additionally, we proposed that TolQRA machines, powered by the PMF, pull TolB away from Pal thereby releasing it to bind to the cell wall. This hypothesis is supported by the structural similarities between the TolA-TolB complex and that of another PMF-driven machine, TonB, which binds to TonB-dependent transporters [5,10]. TonB, powered by its inner membrane ExbB-ExbD stator complex, displaces the plug domains of outer membrane transporters in order to import bound nutrient substrates [11]. We have speculated that the displaced plug domains may also be brought through holes in the peptidoglycan inner order to activate sigma factors in the inner membrane [5,12]. Thus, in dividing cells, in which TolQRA is localised to the septum as part of the division machinery [7], Pal would be deposited at the septum to fulfil its role in outer membrane constriction.

While this conceptual model, which we termed *mobilisation-and-capture* (M&C), rationalises aspects of Pal behaviour in the outer membrane, such as accumulation at the septum of dividing cells, it is not clear that it can explain the observed mobility changes relative to non-dividing cells. Moreover, Pal mobility in *tol* mutants tested previously was found to be essentially identical to that in non-dividing cells, implying that the Tol-Pal system is effectively inactive in non-dividing cells. Here, to resolve these apparent contradictions, we develop a mathematical model of the Tol-Pal system, based on the physical properties of diffusion and

spatially-localised dissociation, that shows, quantitatively, how TolB mobilises Pal only during division and transports it to division septa. It not only explains Pal mobility in wild-type and *tol* mutants, but also successfully predicts differences in TolB mobility as well as the effect of varying TolA levels. Overall, our model explains how Pal is redistributed to division septa to coordinate outer membrane invagination, one of the final steps in Gram-negative cell division. Furthermore, as the fundamental properties on which the model relies are physical consequences of localised dissociation, the insights gained may be applicable to other systems enclosed by double membranes such as mitochondria and chloroplasts.

## Results

### Localised transport is less efficient than homogenous transport

We have previously used our SpatialFRAP method to measure the mobility of Pal across the length of the cell, finding that in dividing cells Pal mobility away from the septum is greater than in non-dividing cells [5]. This mobility was found to be dependent on the Tol system. However, while the spatial localisation of TolQRA complexes to the division septum and their hypothesised deposition of Pal at that location may explain the spatial variation in $D_{eff}$, it is not clear why overall Pal mobility would be higher in dividing cells (the orange curve in Fig 1D is entirely above the blue curve), especially as we found no evidence that any of the components of the Tol system are relatively more abundant in dividing cells (S1 Fig).

However, the overall increase in Pal mobility suggests that there is more TolB in contact with Pal in dividing cells than in non-dividing. This leads us to propose that the cell wall separates the periplasm into inner (inner-membrane proximal) and outer (outer-membrane proximal) compartments and that TolA, as part of the TolQRA complex, pulls TolB from the outer to inner periplasm through holes in the peptidoglycan, powered by the proton motive force. Holes in the cell wall are large enough to accommodate a protein the size of TolB [13]. Once released, TolB diffuses in the inner periplasmic region but can migrate back to the outer layer through the same holes in the peptidoglycan.

To demonstrate how this scheme can lead to more TolB in the outer periplasm of dividing cells, and therefore in contact with Pal, we first developed a simple mathematical model of TolB and its transport across the peptidoglycan layer (Fig 2A). Let $B_{out}(x,t)$ and $B_{in}(x,t)$ denote the concentrations of TolB in the outer and inner periplasm respectively. We assume that TolB is diffuse within these compartments with different diffusion coefficients, $D_{out}$ and $D_{in}$ respectively. The underlying reason for the difference in diffusion coefficients (binding to Pal) is not important for the moment. While transport from the inner to the outer compartment occurs spatially uniformly with rate $\alpha$, transport from the outer to the inner occurs with a potentially spatially varying rate $\beta(x)$ that represents the transport by the TolQRA complexes. In one dimension, we have the following equations

$$\frac{\partial B_{out}}{\partial t} = D_{out}\frac{\partial^2 B_{out}}{\partial x^2} + \alpha B_{in} - \beta(x)B_{out} \tag{1A}$$

$$\frac{\partial B_{in}}{\partial t} = D_{in}\frac{\partial^2 B_{in}}{\partial x^2} - \alpha B_{in} + \beta(x)B_{out} \tag{1B}$$

which we take over a spatial domain [-L/2, L/2] with reflective boundary conditions. Note that the total TolB concentration is conserved i.e. we do not consider production or degradation.

We consider two different cases (Fig 2A): homogeneous transport where $\beta(x) = \frac{\beta_0}{L}$ is a spatial constant and localised transport at the centre position $x = 0$ where $\beta(x) = \beta_0\delta(x)$. Note the total number of TolQRA complexes is the same in both cases i.e. $\int \beta(x)\,dx = \beta_0$. Solving these

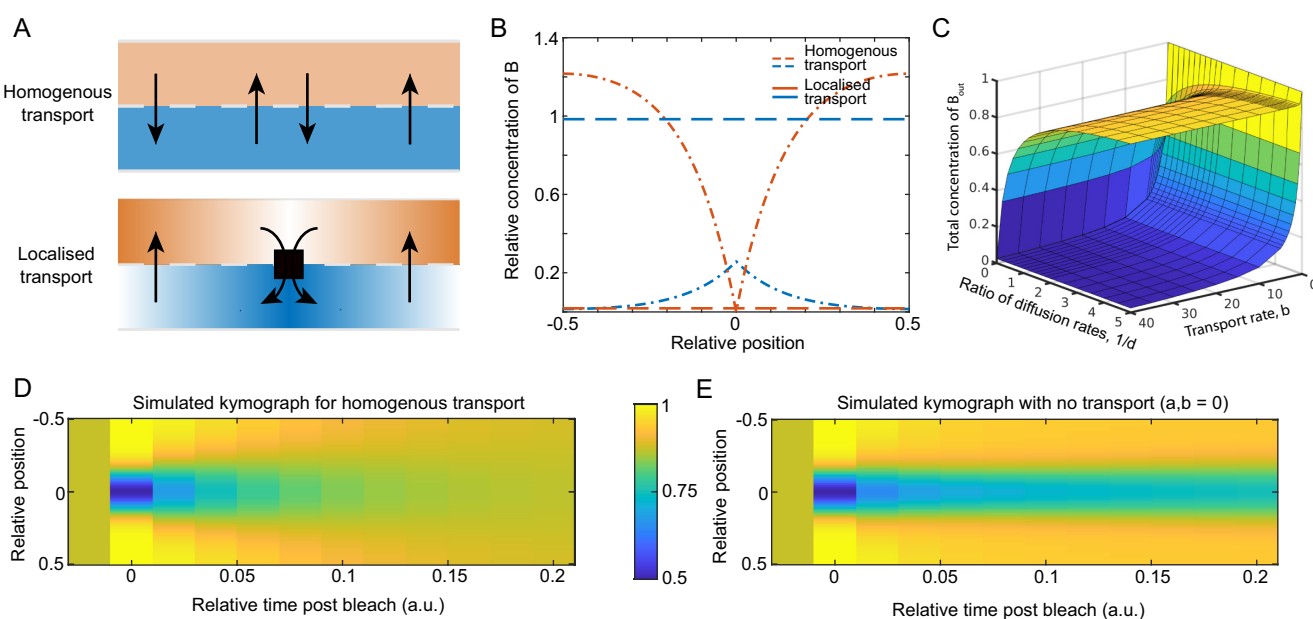

**Fig 2. Localised transport is less efficient than homogenous transport.** (A) We compare two schemes for protein transport from the outer to inner periplasm: spatially uniformly across the length of the cell (homogeneous transport; top) and only at the centre of the cell (localised transport; bottom). In both cases the transport in the opposite direction, from inner to outer, is spatially uniform. Here, the blue and orange distinguish the inner and outer periplasm respectively. (B) Example line profiles of the concentration in the inner (blue) and outer (orange) periplasm for homogenous transport (dashed) and localised (dot-dashed) transport. The average transport rate is the same in both cases. Only the localisation of the transport differs. Note that there is more protein in the outer periplasm for localised transport. (C) A surface plot showing the total concentration of $B_{out}(x,t)$, in the case of homogenous transport (lower surface) and localised transport (upper surface), as the dimensionless parameters $d = D_{out}/D_{in}$ and $b = \beta_0/\alpha L$ are varied. Here $a = \alpha L^2/D_{in}$. The parameters in (B) are $a = 50$, $b = 100$, and $d = 0.2$. See also S2 Fig. (D) Kymographs of simulated FRAP experiments with (left) and without (right) homogeneous transport. Both cases have the same steady state with the same concentration of molecules in the inner and outer periplasm. Note that the kymograph with transport returns to the initial steady state faster, showing how the transport-induced reshuffling of the slower sub-population increases the effective rate of diffusion of the entire population. Note that the system with no transport will eventually return to the initial steady state. In both kymographs $d = 0.02$, and with homogenous transport $a = 50$, and $b = 1$.

equations first for the homogenous case, we find, as expected, uniform steady-state profiles of TolB in both compartments (Fig 2B, dashed lines). For localised transport however the solution is hyperbolic in shape on either side of the centre position (Fig 2B, dot-dashed lines) due to the combination of diffusion and localised transport (see S1 File).

Interestingly, we find that the total amount of TolB in the outer periplasm is greater for localised transport than for homogeneous transport (compare orange lines in Fig 2B) i.e. TolQRA complexes are overall less efficient at transporting TolB when they are localised than when they are uniformly distributed. This can be understood in terms of the time it takes for a TolB molecule that has just migrated from the inner to outer compartments to be transported back by a TolQRA complex. In the localised case, the molecule may have to diffuse as much as half a cell length, whereas in the homogenous case it is more likely to find a TolQRA complex in its vicinity. This result holds for any choice of parameters, but notably is greater the slower the diffusion of TolB in the outer compartment (see S2 Fig).

In our previous study, we found that preventing TolB from interacting with Pal (via two point mutations) abrogates the increased-mobility effect, with dividing cells then having the same uniform effective diffusion coefficient as wild-type non-dividing cells (Fig 1D). This suggests that TolB does not interact with Pal in non-dividing cells. In the context of our inner/outer periplasm hypothesis, the implication is that in the homogenous case most TolB is in the inner periplasm. In our model above, this occurs when $b = \frac{\beta_0}{\alpha L} \gg 1$ i.e. when transport of TolB from the outer to the inner compartment (by TolQRA) is sufficiently greater than its transport

in the opposite direction. Importantly, having the majority of TolB in the inner compartment in non-dividing cells is not mutually exclusive with having most of TolB in the outer compartment in dividing cells (in which transport is localised), which requires $D_{out}$ to be sufficiently small (Fig 2C), i.e. these two criteria can be independently satisfied. We will address biological relevant parameter values below.

## Exchange between differently diffusing states affects overall mobility

Another challenging result to explain was our finding that in the absence of TolA (Fig 1D), or under disruption of its coupling to the PMF (using TolA H22A) [5], Pal mobility was found to be indistinguishable from that in non-dividing cells, just as for the TolB disruption discussed above. The naive expectation is that in the absence of TolA, TolB would not be efficiently transported from the outer to the inner periplasm and therefore there should be more TolB available in the outer periplasm to mobilise Pal. However, this neglects the effect of exchange on global measures of mobility.

Consider again the simple model of TolB transport introduced above. Suppose we were to measure TolB mobility using a population-based technique such as FRAP. Using the model, we could simulate this in both the presence and absence of homogenous transport, keeping the same steady-state concentration in each compartment across the two cases. Interestingly, we found that the recovery after simulated bleaching is faster in the presence of transport than without (Fig 2D and 2E), despite the fact that the average diffusion coefficient of the entire population is the same in both cases. This is because transport between the two compartments results in a faster effective recovery of the more slowly diffusing population. This is a general result and holds for any system having exchange between differently diffusing states, a very common scenario in biology e.g. DNA or membrane binding proteins. While somewhat obvious in hindsight, we believe that it is an underappreciated observation.

What is the relevance to Pal mobility in the *tolA* deletion strain? While there may be more TolB in the outer periplasm in this mutant, the resulting TolB-Pal complexes are not being continuously disassembled by the action of TolQRA. In wild-type cells the continuous turnover of TolB ensures that the particular subset of Pal molecules mobilised by TolB is continuously changing. In the absence of TolA, the exchange between the immobile (PG-bound) and mobile (in Pal-TolB complexes) states of Pal is greatly reduced. As a result, any increase in TolB levels in the outer periplasm in a *tolA* mutant may be ineffective, on the population level, at mobilising Pal, which is ten times more abundant [14]. We will return to this point below after developing a more complete model of the Tol-Pal system.

## A minimal model of the Tol-Pal system

Our minimal model of the Tol-Pal system is built on the fundamental premise, introduced above, that the periplasm is separated into two compartments, inner and outer. Further, we propose that TolA, which, as part of TolQRA inner membrane complexes, extends through holes in the PG layer, binds to TolB in the outer periplasm and pulls it into the inner periplasm, dissembling TolB-Pal complexes and releasing Pal in the process (Fig 3A). The only difference between dividing and non-dividing cells is the distribution of TolQRA complexes within the cell: homogeneously distributed in non-dividing cells; localised to the septum in dividing cells (Fig 3B). It should be noted that the molecular basis for TolQRA localisation to the septum is not currently understood but recent work has identified a dependence on the FtsWI synthase [15].

Mathematically, the model contains four variables, three of which represent the concentration of one of the different states of Pal. Firstly, Eq (2A) describes the population of Pal in complex with TolB, $C(x,t)$. These complexes can either be actively dissociated by the TolQRA

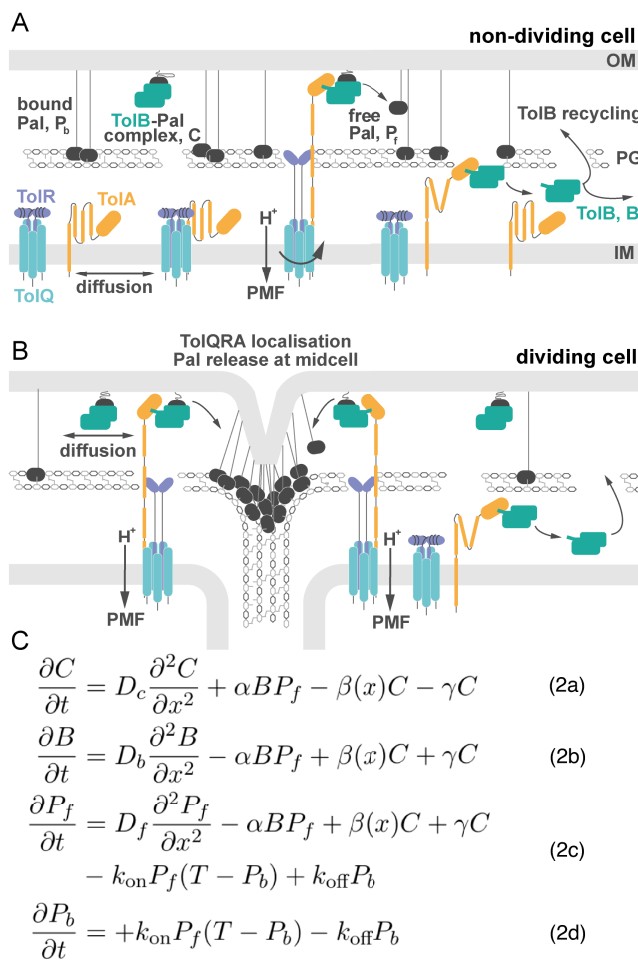

$$\frac{\partial C}{\partial t} = D_c \frac{\partial^2 C}{\partial x^2} + \alpha B P_f - \beta(x)C - \gamma C \qquad (2a)$$

$$\frac{\partial B}{\partial t} = D_b \frac{\partial^2 B}{\partial x^2} - \alpha B P_f + \beta(x)C + \gamma C \qquad (2b)$$

$$\frac{\partial P_f}{\partial t} = D_f \frac{\partial^2 P_f}{\partial x^2} - \alpha B P_f + \beta(x)C + \gamma C$$
$$- k_{\mathrm{on}} P_f (T - P_b) + k_{\mathrm{off}} P_b \qquad (2c)$$

$$\frac{\partial P_b}{\partial t} = + k_{\mathrm{on}} P_f (T - P_b) - k_{\mathrm{off}} P_b \qquad (2d)$$

**Fig 3. A minimal model of the Tol-Pal system.** (A) Non-dividing cell: Here TolQRA is homogeneously distributed throughout the inner membrane. As a result, TolB is efficiently captured by the TolQRA machine and moved back into the inner periplasm. Pal is predominately free of TolB and hence binds to the peptidoglycan layer slowing its diffusion. (B) Dividing cell: TolQRA complexes become localised to the cell septum [7,8] such that TolB transport by TolQRA is localised to the centre of the cell. This leads to less overall transport of TolB, resulting in more TolB in the outer periplasm, where it binds Pal, increasing Pal mobility everywhere except at the septum. In this way TolB can mobilise Pal in dividing cells. (C) System of partial differential equations describing the main components of the Tol-Pal system. See the main text for explanation.

machine at a potentially spatially-dependent rate $\beta(x)$ or they can dissociate independently with rate $\gamma$. In either case the complex is broken down into free TolB, $B(x,t)$, and a transient population of free Pal, $P_f(x,t)$. We assume that the transient state of TolB bound to TolA dissociates rapidly, consistent with their low affinity for each other [4,5]. Therefore, we do not explicitly model this state and instead assume that TolB is immediately returned to the inner periplasm, where it is free to diffuse. TolB can then migrate through holes in the PG layer to the outer periplasm where it can bind to free Pal to again form a TolB-Pal complex. We assume this to occur in one step with rate $\alpha$. This assumption is justified by the relative abundance of Pal molecules (there are ~60,000 Pal and ~6,000 TolB molecules in the periplasm [14]. This is described in Eq (2B) and (2C). The transient free population of Pal is immobilised by binding to the peptidoglycan of the cell wall. This occurs with rate $k_{on}$ but this is limited by the abundance of PG binding sites $T$, which we take to be spatially uniform. This limitation prevents

the over-accumulation of Pal at the septum. The resulting bound population, $P_b(x,t)$, can dissociate at a rate $k_{off}$ to return to the transient free population $P_f(x,t)$ (Eq (2D)). The three mobile species, TolB, TolB-Pal complex and free Pal diffuse with diffusion coefficients $D_b$, $D_c$, and $D_f$ respectively. We model the system in one spatial dimension with reflective boundary conditions. This is consistent with our microscopy (FRAP) approach in which the signal along the long axis of the cell is analysed. The reaction-diffusion equations describing the model are given in Fig 3C. The difference between dividing and non-dividing cells is encoded in the function $\beta(x)$, which represents the distribution of TolQRA complexes. For non-dividing cells, we set it to be a constant $\frac{\beta_0}{L}$, while in dividing we use $\beta(x) = \beta_0 N(0, \sigma^2)$, where $N(0, \sigma^2)$ is the (truncated) normal distribution centered at x = 0, the middle of the cell. Using previous experimental results and estimates we are able to obtain reasonable values for many of the model parameters. This left three unknown parameters, $D_b$, $D_c$ and $\beta_0$ (see Table 1).

To proceed, we first reproduced, using a different microscope, our earlier results by performing FRAP of Pal-mCherry in dividing and non-dividing cells (Fig 4A) and using our SpatialFRAP method to obtain a spatially varying effective diffusion coefficient along the length of the cell (S3 Fig). This gave results very similar to those obtained previously (Fig 1D). Next, we simulated the experimental FRAP procedure using our mathematical model and fitted the results to the corresponding kymographs and effective diffusion coefficients (see Materials and Methods). We found that the model could indeed accurately reproduce the experimental FRAP kymographs and the effective diffusion coefficient (Fig 4A, 4B and 4C). Furthermore, we found the fitted values resulted in TolB being mostly in the inner periplasm in non-dividing cells and mostly in complex with Pal in the outer periplasm in dividing cells (Fig 4D), consistent with the conclusions drawn above using our simple model of TolB transport. These results demonstrate that the transport of TolB across the PG layer, powered by the proton motive force across the inner membrane, can, at least in principle, explain the observed mobilisation of Pal by TolB and its re-distribution across the outer membrane. To test the model further, we next assessed its quantitative predictions.

We have previously seen that Pal mobility in the absence of either functional TolA or TolB is very similar to that of wild-type non-dividing cells (Fig 1B). To test if this is reproduced in the model, we simulated the FRAP procedure in the absence of these proteins but otherwise using the same parameters as used above to fit to the Pal kymographs (Fig 4). Consistent with the experimental measurements, we found that the recovery in both cases to be slow (Fig 5A

**Table 1. Model Parameters.**

| Parameter | Brief description | Value | Source |
|---|---|---|---|
| $D_c$ | Diffusion constant for the TolB-Pal complex | 0.0068 $\mu m^2 s^{-1}$ | Our fitting |
| $D_b$ | Diffusion constant for free TolB | 0.0036 $\mu m^2 s^{-1}$ | Our fitting |
| $D_f$ | Diffusion constant for free Pal | Equal to $D_c$ | We assume that the limiting factor for diffusion is the embedding of Pal's lipoylated domain in the outer membrane, similar to how the mobility of membrane proteins is determined by the number of transmembrane domains[19] |
| $\alpha$ | Rate of binding of TolB and Pal | 0.054 $\mu M^{-1} s^{-1}$ | [17] |
| $\beta_0$ | Rate the TolB-Pal complex is pulled apart by TolQRA | 17 $s^{-1}$ | Our fitting |
| $\gamma$ | Rate the TolB-Pal complex dissociates | 0.006 $s^{-1}$ | [17] |
| $k_{on}$ | Rate Pal binds to the peptidoglycan | 0.1 $\mu M^{-1} s^{-1}$ | Estimate |
| $k_{off}$ | Rate Pal unbinds peptidoglycan | 1.0 $s^{-1}$ | Estimate |
| T | Concentration of peptidoglycan binding sites | 320 $\mu M$ | Found from the height difference of the valley of the effective diffusion coefficient |
| $\sigma$ | Standard deviation of the truncated normal distribution for the shape of TolQRA in dividing cells | 0.08 | Found from fitting to the shape of the valley in the effective diffusion coefficient |

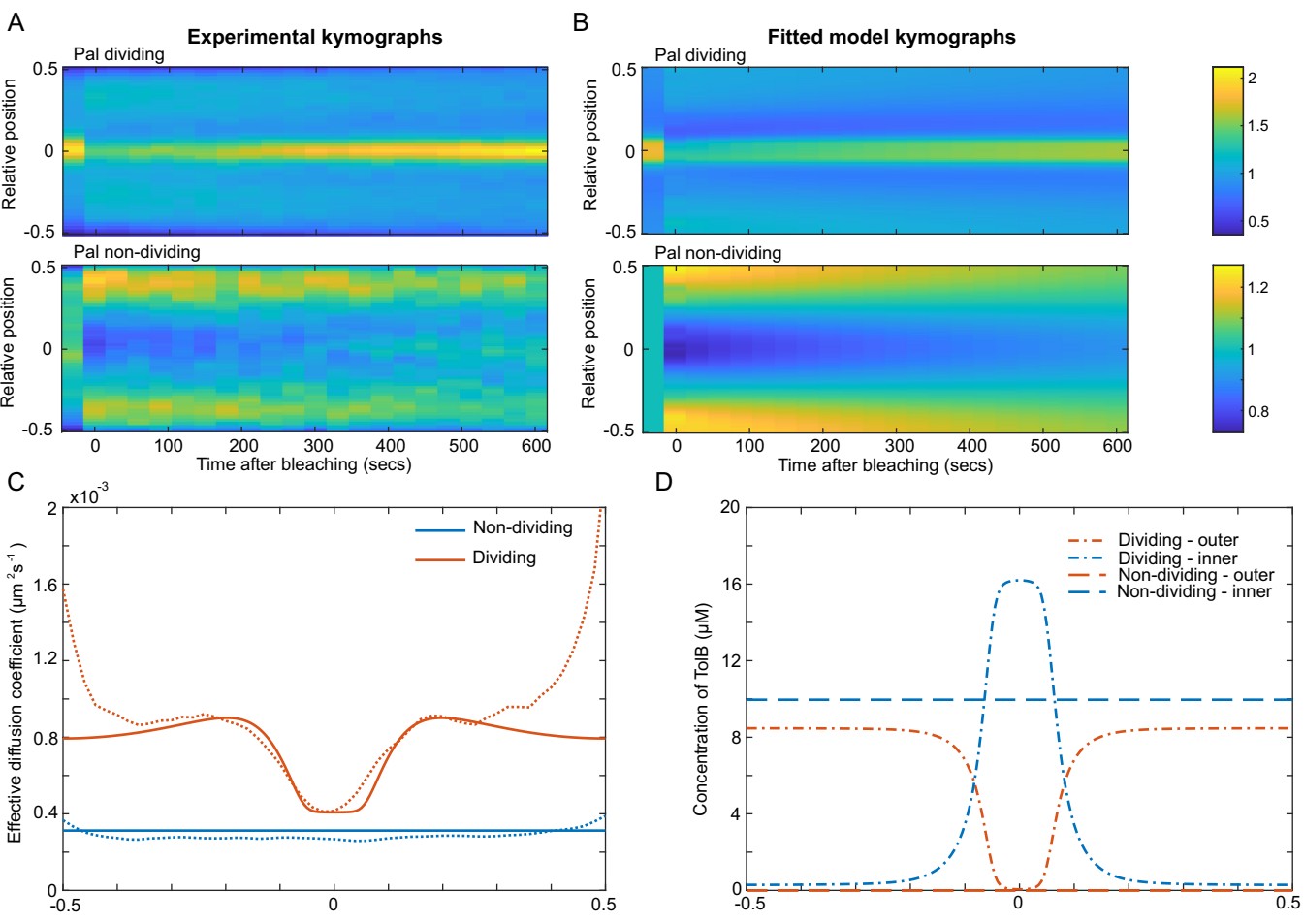

**Fig 4. Fitting the model to experimental Pal SpatialFRAP.** (A) Kymographs of FRAP of Pal-mCherry performed averaged over 30 dividing (top) and non-dividing (bottom) cells. See also S3 Fig. (B) Simulated kymographs from the mathematical model showing the results for the best fit to the experimental data in (A). Colour bars are for both (A) and (B) and show either the normalised fluorescence (experimental data) or the normalised concentration (simulations). (C) Comparison between the effective diffusion coefficient computed using the SpatialFRAP method from the average experimental kymographs in (A) (dashed lines) and the best fit of the model (solid lines) for Pal in non-dividing (blue lines) and dividing (orange lines) cells. (D) The distribution of TolB in the inner (blue lines) and outer periplasm (orange lines) for dividing (dot-dash lines) and non-dividing (dashed lines) cells for the best fit shown in (B). Note the greater concentration of TolB in the outer periplasm of dividing cells (compare orange lines).

and 5B) with an effective diffusion coefficient very similar to that in non-dividing wild-type cells (S4 Fig). The reasons for these results are clear. In the absence of TolB, Pal can no longer form mobile complexes with TolB and is not deposited at the septum. This results in Pal mobility being similar to that in non-dividing cells since in that cell-type TolB is almost entirely in the inner periplasm where it cannot interact with Pal. In Δ*tolA* cells, TolB is present in the outer periplasm but due to the lack of TolA, the turnover of Pal-TolB complexes is reduced. As we have seen above (Fig 2D and 2E), this lack of exchange results in slower recovery of the Pal signal than would be expected otherwise and we again see recovery more similar to that of non-dividing than of dividing cells.

## Prediction of TolB mobility

It has previously been shown that TolB exhibits mild accumulation at the septum of dividing cells [8] (Fig 5C). Analysing the mathematical properties of our model (see S1 File), we find

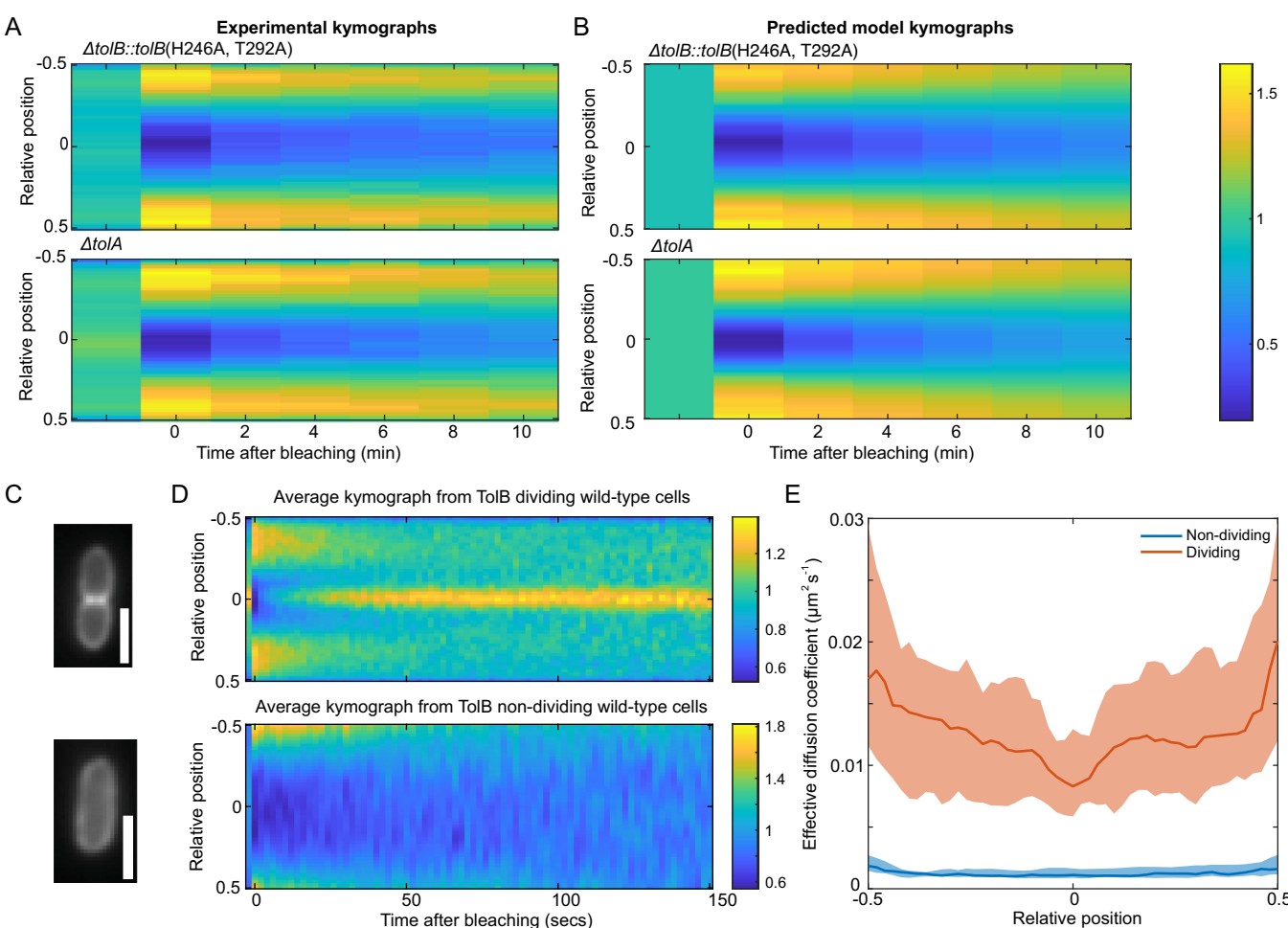

**Fig 5. The model for the Tol-Pal system can predict the effects of mutants and TolB mobility.** (A) Kymographs of FRAP of Pal-mCherry averaged over 30 *ΔtolB::tolB*(H246A, T292A) (top) and *ΔtolA* (bottom) mutant cells. Data from reference [5]. (B) Predicted kymographs from the model using the fitting in Fig 4. See also S4 Fig. Colour bar is for both (A) and (B) and shows either the normalised fluorescence (experimental data) or normalised concentration (simulations). (C) Top, example cell showing the localisation of TolB-mCherry in dividing cells. Bottom, example cell showing the localisation of TolB-mCherry in non-dividing cells. In dividing cells, TolB is seen to be localised to the centre of the cells whereas in non-dividing cells it is homogeneously distributed, similarly to the distribution of Pal. Scale bar, 1 μm. (D) Experimental FRAP kymographs of TolB-mCherry showing the average from 30 dividing (top) and non-dividing (bottom) cells. Colour bars show normalised fluorescence. (E) Effective diffusion coefficient of TolB-mCherry in individual non-dividing cells (blue line) and dividing cells (orange line). In dividing cells, the diffusion coefficient is higher than in non-dividing cells. Based on the model, most TolB in dividing cells is in the outer periplasm whereas in non-dividing the majority is in the inner periplasm. This suggests that the diffusion coefficient of TolB in the outer periplasm is faster than that of TolB in the inner periplasm. Shown is the median of approximately 30 cells as a function of relative long axis position with 95% confidence intervals.

that this can only occur if the diffusion coefficient of TolB in the inner periplasm is lower than that of TolB in the outer periplasm (which is in complex with Pal). This is unexpected from a biochemical viewpoint since TolB-Pal complexes are tethered to the outer membrane via the lipoylated domain of Pal and would therefore be expected to diffuse more slowly than free TolB.

To test this prediction of the model, we performed FRAP on cells carrying a TolB-mCherry fusion (S5 Fig) and used our SpatialFRAP method to assess its mobility in both dividing and non-dividing cells. We found that TolB-mCherry in dividing cells displays a significantly higher mobility than in non-dividing cells (Fig 5D and 5E) consistent with the output of the model (S5 Fig). Since the model predicts that the majority of TolB is in the inner periplasm in non-dividing cells but in the outer periplasm in dividing cells, the effective diffusion coefficient

in each cell type reflects the diffusion coefficient of the free and Pal-bound states, respectively. Therefore, the increased $D_{eff}$ of dividing cells is consistent with the model prediction that the mobility of TolB-Pal complexes is greater than that of free TolB. We hypothesise that this may be due to interaction between TolB and TolA in the inner periplasm following PMF activation. This is a reasonable assumption given that in the absence of the PMF TolB forms a higher affinity complex with TolA than it does when bound to Pal [4].

## Sequestration of TolB abrogates Pal mobilisation in dividing cells

Our model is based on the mobilisation of Pal due to its binding to TolB. We therefore reasoned that decreasing the amount of TolB available to bind Pal would result in a corresponding decrease of Pal mobility. Such a decrease could be achieved by overexpressing TolA, which is inserted into the inner membrane independently of TolQ and TolR; and is recruited to the septum separately from the stator [7]. The resulting excess of TolA in the inner membrane would compete for TolB and sequester it away from Pal. Since TolQ and TolR are not under inducible control, the number of functional TolQRA machines should be largely unaffected.

To test this prediction we used an arabinose inducible promoter to overexpress TolA (S6 Fig) and measured Pal mobility changes using our SpatialFRAP technique. We found that Pal mobility was reduced by TolA induction (Fig 6A). At the highest level measured (0.2% arabinose), Pal recovery after bleaching was significantly slower than with no inducer, with the focus of Pal at the septum still not having recovered after 10 min. (Fig 6B), resulting in a much lower $D_{eff}$ (Fig 6A). Importantly, our model was able to reproduce this result. We found that reducing TolB levels in the model resulted in a similar reduction of Pal mobility as observed experimentally (S6 Fig). Notwithstanding the molecular basis for the inhibitory effect of TolA overexpression, these data reinforce the supposition that modulating levels of TolA, which does not interact with Pal [4], nevertheless influences Pal dynamics via its effect on TolB. Overall, our quantitative model is able to explain how TolQRA, in conjunction with TolB, induces the relocalisation of Pal to the division septum and successfully recapitulates the behaviour of the system under several perturbations.

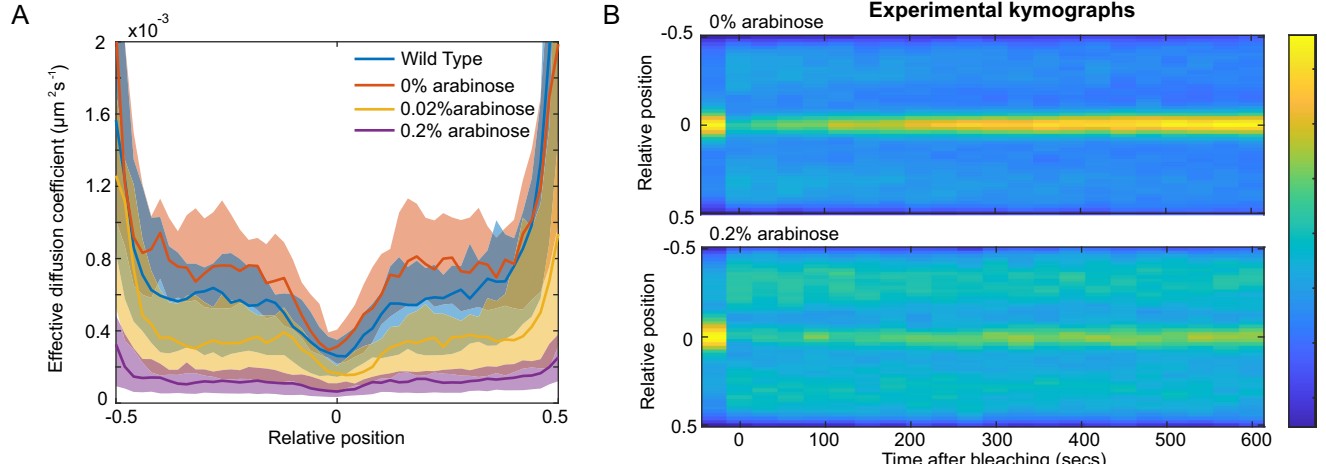

**Fig 6. Increasing the concentration of TolA lowers the effective diffusion coefficient of Pal.** (A) The effective diffusion coefficient of Pal-mCherry in dividing cells for wild type and at different levels of *tolA* expression. As the concentration of arabinose increases the effective diffusion coefficient of Pal decreases. Shown is the median and 95% confidence intervals from approximately 30 cells for each case. (B) Averaged kymographs of FRAP on Pal-mCherry (data as in (A)) for 0% (top) and 0.2% (bottom) arabinose induction. While the septal signal recovers fully by 10 min at 0% arabinose, it only partially recovers for 0.2% arabinose. See also S6 Fig.

## Discussion

In this work we show how the active transport of a protein *across* the bacterial cell wall can be leveraged to power the redistribution of another protein *along* the outer membrane. We focused on Pal, a slowly-diffusing bacterial lipoprotein found in many Gram-negative bacteria. The role of Pal is to tether the outer membrane to the peptidoglycan of the cell wall, particularly at the division site, where it accumulates late in the division process due to the action of the Tol system [7]. This facilitates the invagination of the outer membrane during cell division, likely by a combination of tethering and cell wall remodelling [16]. However, the low mobility of Pal means that its recruitment cannot occur through a canonical 'diffusion and capture' mechanism. Furthermore, the periplasm is not energised so the energy for an active mechanism must come from the proton motive force across the inner membrane.

The slow mobility of Pal is attributable to its tethering between the outer membrane (via its lipoylated domain) and the cell wall (via binding to peptidoglycan). However, binding to PG is prevented by complex formation with the periplasmic protein TolB. Thus, we previously proposed that Pal exists primarily in two states: an essentially immobile state bound to PG and as part of mobile TolB-Pal complexes. How are these mobile complexes used to bring Pal to the division septum? Our model is that the inner membrane TolQRA complexes, which are recruited independently to the septum as part of the divisome, pull TolB through holes in the peptidoglycan layer from the outer to inner periplasm, dissociating it from Pal in the process, and eventually returning through the same holes to repeat the cycle in a manner akin to a conveyor belt. In this way, Pal is mobilised from across the cell and transported to the septum to play its role in invaginating the OM. However, it was not at all clear that this scheme could explain why Pal is mobilised only in dividing cells and why its mobilisation requires functioning TolQRA machines.

Here, we have used mathematical modelling to show that 'mobilisation and capture' model can indeed explain all the measurements of Pal mobility. In dividing cells, the TolB 'conveyor belt' results in greater Pal mobility away from, rather than at, the septum (Fig 4C). This mobility is greater than in non-dividing cells because localised transport is less efficient than homogenous transport (Fig 2). As a result, dividing cells have, in total, more TolB in the outer periplasm than in non-dividing cells and therefore greater Pal mobility (Fig 4C). Indeed, the fact that Pal mobility in cells lacking TolB is indistinguishable from that in wild-type non-dividing cells indicates that the majority of TolB in non-dividing cells is in the inner periplasm where it cannot interact with Pal. This is also consistent with active transport across the cell wall in one direction only. In cells lacking TolA, the active transport of TolB is disrupted. However, this does not lead to an increase in Pal mobility because without recycling by the TolQRA machine, TolB-Pal complexes are relatively stable (with an *in vitro* dissociation half-time of 2 min [17]) and therefore, within the timeframe of the FRAP experiment, mobilise a smaller fraction of Pal than in wild-type dividing cells.

Our mathematical formulation of the model (Fig 3) was able to quantitatively reproduce these results using only the wild-type data for parameter estimation (Figs 4, 5A and 5B). It also predicted that TolB mobility would be greater in dividing relative to non-dividing cells, which we confirmed experimentally (Fig 5). We also examined the effect of TolA over-expression. We hypothesised that this would titrate TolB away from Pal and therefore reduce Pal mobility in dividing cells. We confirmed this to be the case, finding a clear negative relationship between increasing TolA levels and a lower effective diffusion coefficient of Pal. In the future, the model could be tested further by modulating TolB levels. While we have found that plasmid-expressed TolB is toxic for cells, this might be surmountable by inducing TolB expression from the chromosome. This would allow us to test the model prediction of a positive relationship between TolB levels and Pal mobility.

Collectively, our data and modelling support a 'mobilisation and capture' mechanism for the mobilisation and redistribution of Pal to the division site by the action of the Tol system. The mechanism is efficient in that it naturally leads to Pal mobilisation only in dividing cells (in which TolQRA becomes localised to the septum). Furthermore, while the capture of TolB-Pal complexes by TolQRA machines is via 'diffusion-and-capture', the regulation of mobilisation makes it a distinct mechanism. More generally, our model demonstrates how an immobile protein can be actively translocated across the outer membrane despite the latter's non-energised nature.

A core component of our model of the Tol-Pal system is the active dissociation of TolB-Pal complexes by the TolQRA machines. This was based on the structural homologies to TonB dependent transporters, molecular dynamics simulations of the dissociation and the fact that TolA coupling to the PMF is required for the redistribution of Pal [5]. Secondly, our model assumes separation of the periplasm into inner and outer compartments (at least from the perspective of TolB). As well as the homologies to TonB dependent transporters, this is supported by our previous result that TolA undergoes a PMF-dependent extension through the periplasm [18]. However, this separation is not a strict requirement of the model. The critical ingredient is simply that the dissociation of TolB-Pal complexes by TolQRA machines is spatially localised to the septum. If the cell wall is relatively porous for TolB or if TolB is not pulled through the cell wall as it is dissociated from Pal, adapting the model would only require the re-interpretation of all the model variables as being in the same compartment. The equations would remain unchanged. As a result, the mechanism that we propose here may be applicable to any organelle with two membranes, such as mitochondria and chloroplasts, wherein outer membrane protein redistribution needs to be coordinated with inner membrane invagination.

## Materials and methods

### SpatialFRAP

Finding the effective diffusion coefficient from the kymographs was performed as in [5]. In that work we considered the molecule of interest (Pal) exchanges between mobile and immobile states according to its binding to peptidoglycan. Then, assuming we are in the effective diffusion regime, we found the total Pal concentration evolves according to a Fokker-Planck equation. Here, we generalise this result. We directly model the inhomogeneity in the fluorescence profiles of Pal-mCherry and TolB-mCherry as being due to an inhomogeneous diffusion coefficient. This then allows us to move directly to the Fokker-Planck equation describing the diffusion of the total concentration, c, of the population throughout the cell:

$$\frac{\partial c}{\partial t} = \frac{\partial^2}{\partial x^2}\left(D_{eff}(x)c\right).$$

This equation has a non-uniform steady state described by the spatially varying diffusion coefficient $D_{eff}(x)$. We can obtain $D_{eff}(x)$ by fitting the line profiles during a FRAP experiment to a numerical of this equation. We find the mean signal along the length of the cell on each frame, then remove both the first and the last two pixels in order to remove most of the effects due to the lower signal at the poles. We then normalise the sum of the signal across each frame to 1. We then take the pre-bleach frame as the equilibrium profile. The equation above, with reflective boundary conditions, has an equilibrium profile that is up to a constant factor, the reciprocal of $D_{eff}(x)$. The pre-bleach frame is therefore taken to specify $D_{eff}(x)$ up to a constant, we then determine the proportionality constant by fitting the solution to the above equation with the first post-bleach frame as the initial condition. Including the first post-bleach frame, we then fit the output to the data from the post-bleach frames of the experimental data. Time intervals are taken as they are specified in the experimental data (30 s/2 min for Pal and 2 s for

TolB). We used the *pdepe* solver in MATLAB to solve the equation for c and the function *immse* to calculate the mean square error between the simulated and experimental data (the value was multiplied by 100,000 to avoid numerical issues with small numbers). The fitting was performed by the *patternsearch* function of the Global optimization toolbox. The initial guess was a diffusion constant of $10^{-3}$ μm$^2$s$^{-1}$ converted into units of pixels. Applying this procedure we obtain a spatially varying effective diffusion constant for each cell. The scripts for the SpatialFRAP technique are available at https://github.com/smury/SpatialFRAP.

## Fitting the model to Pal FRAP data

In order to find the values for unknown parameters $D_b$, $D_c$, and beta$_0$ the model is fitted to both the kymographs and the effective diffusion coefficients for Pal in dividing and non-dividing cells. $D_c$ and $D_b$ are fitted for using the parameters $a$ and $b$ which are related to $D_c$ and $D_b$ by:

$$a = D_c - D_b \text{ and } b = \frac{D_c}{D_b}.$$

Then enforcing the bounds $a > 0$, and $b > 1$ ensures that $D_c > D_b$ as we have proven is necessary to give a peak in the total concentration of TolB. During the fitting procedure $D_f$ is chosen as being equal to $D_c$ and all the remaining parameters are fixed as: $\alpha = 0.054$ μM$^{-1}$s$^{-1}$, $\gamma = 0.006$ s$^{-1}$, $k_{on} = 0.1$ μM$^{-1}$s$^{-1}$, $k_{off} = 1.0$ s$^{-1}$, $T = 320$ μM, and $\sigma = 0.08$. All simulations are run using reflective boundary conditions and over the domain $x \in [-L/2, L/2]$ where the value of $L$ is taken as the average length of the cells from the respective experimental data. For dividing cells $L = 4.0$ μm and for nondividing cells $L = 2.8$ μm.

First, the kymographs are simulated computationally. This is performed using the MATLAB solver *pdepe*. For every parameter choice, the model is first run for a sufficiently long time such that the steady state is reached. This gives the first bre-bleach frame of the kymographs. Then a second simulation is run for both a total population and a bleached population such that we have the set of equations:

$$\frac{\partial C_T}{\partial t} = D_c \frac{\partial^2 C_T}{\partial x^2} + \alpha B_T P_{fT} - \beta(x)C_T - \gamma C_T$$

$$\frac{\partial B_T}{\partial t} = D_b \frac{\partial^2 B_T}{\partial x^2} - \alpha B_T P_{fT} + \beta(x)C_T + \gamma C_T$$

$$\frac{\partial P_{fT}}{\partial t} = D_f \frac{\partial^2 P_{fT}}{\partial x^2} - \alpha B_T P_{fT} + \beta(x)C_T + \gamma C_T - k_{on}P_{fT}(T - P_{bT}) + k_{off}P_{bT}$$

$$\frac{\partial P_{bT}}{\partial t} = k_{on}P_{fT}(T - P_{bT}) - k_{off}P_{bT}$$

$$\frac{\partial C_v}{\partial t} = D_c \frac{\partial^2 C_v}{\partial x^2} + \alpha B_T P_{fv} - \beta(x)C_v - \gamma C_v$$

$$\frac{\partial P_{fv}}{\partial t} = D_f \frac{\partial^2 P_{fv}}{\partial x^2} - \alpha B_T P_{fv} + \beta(x)C_v + \gamma C_v - k_{on}P_{fv}(T - P_{bv}) + k_{off}P_{bv}$$

$$\frac{\partial P_{bv}}{\partial t} = k_{on}P_{fv}(T - P_{bv}) - k_{off}P_{bv}$$

where the subscript T represents the total population and the subscript v corresponds to only the visible population, such that the first four equations are for the total population and the last three for the visible population that remains after bleaching. The initial conditions for the total population are taken as the steady state obtained previously. We obtain a bleaching function from the experimental data by dividing the pre-bleach and post-bleach line profiles. This bleaching function is then applied to the simulated steady state to obtain the concentration profile of the visible population (after bleaching). The coupled system is then solved numerically for the same total time as used in the experimental kymograph procedure. This gives the remainder of the simulated kymograph.

The effective diffusion coefficient is then found from the simulated kymograph using the SpatialFRAP method, in exactly the same way as for the experimental data.

The MATLAB function *immse* is used to calculate the mean square error (the cost function) between the simulated and experimental data for both the kymographs and effective diffusion coefficient. The resultant values are added with weights of $10^5$ (kymographs) and $10^7$ (effective diffusion coefficients) in order to avoid numerical issues with small numbers and to evenly weight the two different procedures. This cost function was minimised over the described parameters using the *patternsearch* function from MATLAB's Global Optimization toolbox.

The Pal $\Delta tolA$ and $\Delta tolB$::$tolB$(H246A,T292A) mutants predicted kymographs are generated following the same procedure described for Pal in dividing and non-dividing cells, only using the best fit values found from the Pal fitting. We produce the TolB predicted kymographs in a similar manner. Here, the TolB-Pal complex and the TolB in the inner periplasm are instead bleached in order to form a set of six equations for both the total concentrations and the visible concentrations. Numerically solving this system then follows from the same method as used above. The Matlab scripts are available at https://github.com/lconnolley/Tol-Pal-model.

## Strain and plasmids construction

Strain JSCK9 (*tolB*::*tolB-GGGGS-mCherry*) was constructed using scarless allelic exchange based on λ-red recombination and I-SceI digestion [20]. The following four fragments were amplified: 1) last 641 bp of *tolB*, 2) mCherry, 3) I-SceI-flanked spectinomycin resistance gene 4) 300 bp region downstream of *tolB* with the last 50 bp of the upstream fragment cloned at its 5' end. Those fragments were assembled using NEBuilder HiFi DNA Assembly Master Mix (#E2621S) according to the manufactures' instruction to make pJS50 plasmid. The insert was excised from pJS50 using EcoRI and HindIII restriction enzymes and electroporated into the BW25113 strain carrying the pREDTKI helper plasmid. The correct insertion of mCherry was confirmed by colony PCR and spcR marker was excised using I-SceI and homologous recombination. The final sequence of the JSCK9 strain was confirmed by Sanger sequencing.

Plasmid pJS61 was constructed by ligating PCR-amplified EcoRI/SalI digest of GFP-TolA from pNP4 into pBAD24.

The list of strains and plasmids is found in Table 2. The primers used are list in Table 3.

## Cell preparation for live microscopy

M9-glucose (2 mM MgSO4, 0.1 mM CaCl2, 0.4% (w/v) d-glucose) cultures were grown overnight, diluted in fresh medium with appropriate antibiotics, and arabinose for strain expressing TolA from pJS61 plasmid, and grown at 37˚C to OD600 0.3. Aliquot of 0.5 ml were centrifuged at 7000 × g for 3 min, resuspended in 50μl of fresh media. Five microliters of cells were pipetted onto M9 agar pad, allowed to dry and sealed with a clean coverslip.

**Table 2. Strains and plasmids used in this study.**

| Strain | Genotype | Reference |
|---|---|---|
| BW25113 | F-, Δ(araD-araB)567 Δ(rhaD-rhaB)568 ΔlacZ4787 (::rrnB-3) hsdR514 rph-1 | [21] |
| JW5100-1 | F-, Δ(araD-araB)567, ΔlacZ4787(::rrnB-3), ΔtolB789::kan, λ-, rph-1, Δ(rhaD-rhaB)568, hsdR514 | [22] |
| RKCK5 | tolA::GFP(ΔMunI)-tolA | [18] |
| RKCK8 | pal::pal-GGGGS-mCherry(ΔAgeI)(ΔEcoRI)-kan | [5] |
| RKCK10 | ΔtolA pal::pal-GGGGS-mCherry(ΔAgeI)(ΔEcoRI)-kan | [5] |
| JSCK9 | tolB::tolB-GGGGS-mCherry(ΔAgeI)(ΔEcoRI) | This study |
| **Plasmid** | **Genotype** | **Reference** |
| pBAD24 | | [23] |
| pUC19 | | [24] |
| pMDISI | | [20] |
| pREDTKI | | [20] |
| pNP4 | pBR322-GFP-tolA | [7] |
| pJS50 | pUC19-641 bp N-terminus tolB-GGGGS-mCherry-I-SceI-spcR-I-SceI-50 bp N-terminus mCherry-300 bp downstream tolB | This study |
| pJS61 | pBAD24-GFP-t-tolA | This study |

## TIRFM acquisition

Live cells were imaged using an Oxford NanoImager (ONI) superresolution microscope as described in [5]. The distribution of TolB-mCherry fluorescence signal along the *x*-axis of cells was determined by the plot profile function in ImageJ with a line width of 4 points.

## FRAP acquisition

Microscopy was performed on a Zeiss LSM 980 with Airyscan 2 motorised upright laser scanning microscope equipped with DPSS Laser 561 nm (25mW nominal power; 10mW ex fiber) and ×63 oil-immersion objective (Zeiss, NA 1.4) set to 37˚C. Cells were imaged by scanning the laser over a 13.5 × 13.5 μm area with a digital zoom of ×10. The diameter of the pinhole was 0.82 μm. Bleaching was performed using 100% power.

For TolB-mCherry bleaching of the set region of interest (ROI) was performed on 1.5 x 0.5 μm area. Two images were acquired before bleaching and 75 images, with a time interval of 2 s, were recorded after bleaching using an automatic time-course function, laser power 1%

**Table 3. Primers used in this study.**

| Application | Primer sequence (5' to 3') |
|---|---|
| Amplification of last 641 bp of *tolB* | ATGACCATGATTACGCCAAGCTTTGACCTTCGAAAGCGGTCGT (fwr)<br>TCCTCGCCCTTGCTCACCATagaaccaccaccaccCAGATACGGCGACCAGGCAG (rev) |
| Amplification of mCherry | CTGCCTGGTCGCCGTATCTGggtggtggtggttctATGGTGAGCAAGGGCGAGGA (fwr)<br>CAGGGTAATATAATCTCCAGAGGCTACTTGTACAGCTCGTCCA (rev) |
| Amplification of *spcR* | ATGGACGAGCTGTACAAGTAGCCTCTGGAGATTATATTACCCTG (fwr)<br>CCGGTGCTGTGACGACCTTCCGCACCAAGTGGACGATTATTAGGGATAACA (rev) |
| Amplification of 300 bp region downstream of *tolB* with the last 50 bp of the upstream fragment cloned at its 5' end | TCGTCACAGCACCGGCGGCATGGACGAGCTGTACAAGTAGTAATAATTAATTGAATAGTA (fwr)<br>AAACGACGGCCAGTGAATTCTCCAGCATTTGAGCGAAGTC (rev) |
| Amplification of GFP-t-*tolA* | GCAGCAGAATTCGTGTCAAAGGCAACCGAACA (fwr)<br>GCAGCAGTCGACTTACGGTTTGAAGTCCAATG (rev) |

and pixel dwell of 5.76 μs. Images were recorded in 8-bit depth. We observed the formation of polar TolB-mCherry foci in a minority of cells ~1.5hrs after slide preparation. We attributed this to polar expansion of the periplasm due to starvation [25], which we confirmed using a 'mother machine' microfluidic device. We therefore removed these cells from the analysis.

For Pal-mCherry bleaching of the set ROI was performed on 1.5 x 0.2 μm area. One image was acquired before bleaching and 20 images (time interval: 30 s) were recorded after bleaching using an automatic time-course function, laser power 0.5% and pixel dwell of 1.44 μs. Instrument autofocus was used between images in fluorescence mode. Images were recorded in 16-bit depth.

For all fluorescent images, corresponding differential interference contrast (DIC) images were recorded using transmitted light. All images were recorded using Zeiss Zen 2021 software. The distribution of fluorescence along the *x*-axis of cells was determined by the plot profile function in ImageJ with a line width of 12 points.

## Western blotting

Western blotting was carried out as described in [5] using anti-Pal (1:1000), anti-TolB (1:1000) and anti-TolA (1:1000) antibodies.

## Assessing the stability of the E. coli OM

Functionality of fluorescent protein labelled TolA and TolB was examined by growth on LB with added 2% SDS as described in [5].

## Supporting information

**S1 File. Mathematical Analysis.** An analytical analysis of our toy model and a proof of the relationship in the full model between a peak in total TolB and the model diffusion coefficients.
(PDF)

**S1 Fig. TolA and TolB concentration does not correlate with length.** (A) The mean fluorescence of GFP-TolA as a function of the cell length in dividing (orange) and non-dividing (blue) cells. Very weak dependence is found between the mean GFP-TolA fluorescence (a measure of TolA concentration) and length. The Pearson's correlation coefficient is 0.315 with a *p*-value of $1.067 \times 10^{-5}$. (B) The mean fluorescence of TolB-mCherry as a function of the cell length in dividing (orange) and non-dividing (blue) cells. The Pearson's correlation coefficient is -0.204 with a *p*-value of 0.135.
(EPS)

**S2 Fig. Localised transport is less efficient than homogenous transport for a large range of parameter values.** (A) Total concentration of $B_{out}$ as the dimensionless transport rate b is varied. For localised transport, the total concentration in the outer periplasm approaches a non-zero value as the transport rate is increased. In this regime, diffusion becomes the limiting factor. Here, $a = 10$, $d = 1$. (B) Total concentration of $B_{out}$ as the ratio of the diffusion rates, $d$ is varied. In terms of the dimensionful parameters increasing $1/d$ can be achieved by increasing $D_{out}$. Here, $a = 10$, $b = 10$. (C) $D_{in}-D_{out}$ space, the blue shaded region indicates where more than 50% of the TolB is in the outer periplasm, while at the same time, almost all TolB is in the inner periplasm of non-dividing cells ($a = 10$, $\beta_0 = 40$). Note the upper bound on $D_{out}$. (D) Example concentration profiles in the outer (orange) and inner (blue) periplasm for localised transport using a transport function having the shape of a truncated normal distribution (dot-dashed lines) compared to homogeneous transport (dashed lines). This gives a similar result to

that of a point sink: $B_{out}$ is greater for localised transport. See S1 File for further details of this model.
(EPS)

**S3 Fig. Effective diffusion coefficient of Pal.** Effective diffusion coefficient for Pal-mCherry in non-dividing cells (blue line) and dividing cells (orange line) obtained using the Spatial-FRAP method. Shown is the median and 95% confidence intervals over approximately 30 cells. Fig 4C shows the effective diffusion coefficient calculated from the averaged kymograph rather than from the kymographs of individual cells as shown here.
(EPS)

**S4 Fig. The model correctly predicts Pal mobility in two mutants.** (A) Comparison between the effective diffusion coefficient computed using the SpatialFRAP method from the average experimental kymographs (dashed lines) and the best fit of the model (solid lines) for Pal. Wild-type curves are as in Fig 4C. The two mutant cases use the kymographs shown in Fig 5A and 5B. Note that the model correctly predicts both mutants have Pal mobility close to wild-type non-dividing cells. (B) Predicted model concentration profile for TolB in *ΔtolA* cells. The majority of TolB is in the outer periplasm. Despite this the *ΔtolA* mutant still has a low effective diffusion coefficient (yellow line in (A)). This is due to the lack of transport and the resultant recycling of TolB (see Fig 2).
(EPS)

**S5 Fig. The model predicts that TolB mobility is higher in outer periplasm.** (A) Representative Western blots probed with rabbit anti-TolA (1:1000), anti-Pal (1:1000) or anti-TolB (1:1000) antibodies. TolB-mCherry expression is comparable with the wild-type cells and the construct does not impact expression of Pal nor TolA. (B) Wild-type and complemented TolB-mCherry strains show growth on 2% SDS. Representative images from OM stability assays (experiment was repeated three times).(C) Predicted model kymographs for TolB in dividing (top) and non-dividing (bottom) cells based on the best fit of the model to Pal FRAP kymographs (Fig 4). Note the similarity to the experimental FRAP kymographs of TolB-mCherry (Fig 5D). Colour bars show the normalised concentration. (D) Effective diffusion coefficient for TolB in non-dividing (blue line) and dividing (orange line) cells from the predicted model kymographs. It can be seen that the average dividing effective diffusion coefficient is above that of non-dividing as we see experimentally (Fig 5E).
(EPS)

**S6 Fig. TolA overexpression reduces Pal mobility consistent with sequestration of TolB away from Pal.** (A) TolA overexpression is toxic to the cells. Representative images from OM stability assays (n = 3) (B) Representative Western blots probed with rabbit anti-TolA (1:1000), anti-Pal (1:1000) or anti-TolB (1:1000) antibodies. Arabinose titration causes an increase in TolA expression. (C) Fluorescence profiles for GFP-TolA in WT cells (blue line) and for GFP-TolA expressed from a plasmid-based arabinose-inducible promoter for varying concentrations of arabinose (orange, yellow, purple, and green lines). At 0% arabinose concentration there is already significantly more TolA expression than in wild type with the fluorescence signal in the centre of the cell approximately doubling. (D) Same as in (B), showing the fluorescence profiles for greater concentrations of arabinose. (E) Effective diffusion coefficients for wild type and varying arabinose concentrations (dotted lines) and for the model with varying amounts of TolB (solid lines). We hypothesise that excess TolA sequesters TolB, titrating it away from Pal. Decreasing the amount of TolB in the model replicates the result seen experimentally.
(EPS)

## Acknowledgments

We are indebted to Renata Kaminska (Oxford) for invaluable technical support.

## Author Contributions

**Conceptualization:** Seán M. Murray.

**Formal analysis:** Lara Connolley.

**Investigation:** Lara Connolley, Joanna Szczepaniak, Colin Kleanthous, Seán M. Murray.

**Methodology:** Lara Connolley, Joanna Szczepaniak, Colin Kleanthous, Seán M. Murray.

**Project administration:** Seán M. Murray.

**Supervision:** Colin Kleanthous, Seán M. Murray.

**Writing – original draft:** Lara Connolley, Joanna Szczepaniak, Colin Kleanthous, Seán M. Murray.

**Writing – review & editing:** Lara Connolley, Joanna Szczepaniak, Colin Kleanthous, Seán M. Murray.

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
