## [Decision Letter · Decision Letter 0]

19 Oct 2021

Dear Dr. Murray,

Thank you very much for submitting your manuscript "The quantitative basis for the redistribution of immobile bacterial lipoproteins to division septa" for consideration at PLOS Computational Biology.

As with all papers reviewed by the journal, your manuscript was reviewed by members of the editorial board and by several independent reviewers. In light of the reviews (below this email), we would like to invite the resubmission of a significantly-revised version that takes into account the reviewers' comments.

We cannot make any decision about publication until we have seen the revised manuscript and your response to the reviewers' comments. Your revised manuscript is also likely to be sent to reviewers for further evaluation.

Sincerely,

Oleg A Igoshin

Associate Editor

PLOS Computational Biology

Mark Alber

Deputy Editor

PLOS Computational Biology

Reviewer's Responses to Questions

**Comments to the Authors:**

Reviewer #1: Connolley et al. presented a mathematical model to explain the redistribution of immobile bacterial lipoproteins, Pal, to the cell division site. It is known that 1) Pal is immobilized by binding to the peptidoglycan (PG) strands in the periplasm, 2) the Pal-PG and Pal-TolB bindings are mutually exclusive, and 3) Pal-TolB is expected to be more diffusive. Based on these elements, the authors proposed that 1) TolQRA system-mediated proton motive force (PMF) can pull the TolB molecule from Pal-TolB complex, releasing the Pal that will diffuse more rapidly; 2) the TolQRA system exerts the PMF effect everywhere except at the division site. Using this spatial profile of PMF effects in the model, it is not surprisingly that the PG strands at the septum act as a “sink” to trap the free Pal molecules and Pal will become more localized to the division site than other locations.

Conceptually, this proposed scheme is still similar to the typical “diffusion-and-capture” mechanism, except that 1) the PMF is invoked to energize the diffusion of Pal and 2) PMF effects are spatially distributed. Additionally, this work is an extension of the authors’ published work, which has already presented the essence of the model (Szczepaniak et al, Nat Commun 11:1305). Moreover, the authors have not addressed the real mechanistic questions pertaining to the heart of Pal localization pattern, including not limited to: 1) What underlies the spatial regulation of TolQRA system-mediated PMF (low at the septum and high elsewhere)? 2) how does the PMF pull away the TolB from the Pal-TolB complex, given the Pal-TolB complex is mobile? And 3) is this PMF-mediated pulling necessary for the redistribution of Pal to the septum? Instead of pulling, can it be just that TolQRA has a higher binding affinity in the chemical sense that causes some conformational changes in TolB and dissociates the TolB from the Pal-TolB complex?

That being said, while the modeling results seem solid and combine with experiments, which is laudable, they do not present a sufficient conceptual leap nor mechanistic insights. As it currently stands, it seems that this work is more appropriate for a more technical journal. On the other hand, if the authors can improve their mathematical model by faithfully addressing the mechanistic questions like those listed above, then I think this work may have the opportunity to bridge an important gap in understanding how immobile proteins position to certain cellular locations.

Reviewer #2: In the submitted manuscript entitled “The quantitative basis for the redistribution of immobile bacterial lipoproteins to division septa”, the authors proposed a new model to explain the redistribution of slow diffusing lipoprotein Pal to the division septa. Unlike classical “diffusion and capture” model, the authors propose a “mobilization and capture” model for the TolB-Pal complex. The TolQRA is the key player for the model that is homogenously distributed in the non-deviding cells and localized to septum in dividing cells. TolA extends through holes in the PG layer to bind TolB in the outer periplasm and pulls it into the inner periplasm. With this minimal model, the authors can reproduce the diffusion constant of Pal consistent with the experimental results.

Overall, the manuscript is well-written with experimental data and model support. However, I still have some questions regarding the mechanisms and the assumptions.

Major comments

1. In the abstract, it reads “We present a quantitative, mathematical model for Pal relocalisation in which active dissociation of TolB-Pal complexes, powered by the proton motive force across the inner membrane, leads to the net transport of Pal along the outer membrane and its deposition at the division septum.” To my understanding, in the proposed model, the PMF is not directly used for the TolB-Pal complex. This sentence may mislead the readers for the proposed mechanism.

2. In fluid environment, the diffusion constant for small molecule depends on its size. On the membrane, the diffusion constant of membrane protein depends on the number of transmembrane segments. I just wonder how can the diffusing constant of TolB-Pal complex larger than TolB and equal to Pal while TolB-Pal is apparently larger than TolB and Pal alone.

3. One of the critical factor in the model is the TolQRA complex become localized to the cell septum on dividing cells. This is the critical factor that I think it is worthy more explanation/description of the TolQRA localization mechanism.

4. Is there any direct evidence that TolQR using PMF to pull TolB away from Pal? Any direct evidence of TolA can extend through PG holes to bind and pull TolB?

5. It would be useful to have the equation numbers and a paragraph to explains the meaning of these equations to the general readers.

Minor comments

1. In the section “Fitting the model to Pal FRAP data”, line 8, the T=320 “μ”m.

2. Because there are some assumptions in the model, it would be nice to have some outlook regarding the future experiments that can verify the assumptions of the model.

Reviewer #3: Connolley et al present a computational model for the redistribution of the slowly diffusing lipoprotein Pal to the division septum in Escherichia coli. In earlier work, they had found that TolB can increase the mobility of Pal by forming a complex such that Pal no longer binds to peptidoglycan. Based on this and other previous findings they propose the following mechanism for localization of Pal to the division septum: In non-dividing cells TolAQR binds TolB and this prevents TolB from binding to Pal by TolAQR. In dividing cells, TolAQR is directed to the septum such that TolB can increase the diffusion constant of Pal in regions away from the septum. In the present work they study this “mobilisation-and-capture” mechanism by combining theoretical analysis and quantitatively compare their theoretical results with experimental data.

To describe the Pal and TolB dynamics, the authors use a mean-field approach and develop a set of reaction-diffusion equations in one spatial dimension, which the analyze (mostly) numerically. The analysis is sound and experiments and theory are in good agreement, such that the conclusion of this work seems justified. In my opinion this nice piece of work falls into the spectrum of PLOS Computational Biology and should eventually be published in this journal. Before, however, the authors need to address a number of points. Most of them concern the presentation of the results, the quality which is markedly lower than that of the scientific results. I find the presentation sometimes rather convoluted. I give some indications below. Instead of mentioning all the places explicitly, where I think that this is the case, I would invite the authors to ask a colleague to read the manuscript and point out those places that are difficult to access for larger audience. Eventually, however, it’s up to the authors.

1. The authors state that TolB-Pal complexes are actively dissociated by the TolQRA complex. Where does the “activity” enter the equations? Why couldn’t it be just competitive binding that leads to TolB-Pal dissociation by TolQRA? How would the equations change if the system were “passive”?

2. Figure 2.

- panel a: please, indicate what blue and orange colors indicate;

- panel c: there are two surfaces, what do they represent?;

- panel c: what is the total concentration of TolB?;

- panel d: there should be numbers with the color table;

- panel d/e: there is an inconsistency: the caption only mentions d, but a left and a right part;

- panel e: is the system returning to the initial state?

3. Equations for B_out and B_in (after Fig. 2): why are there two different diffusion constants for inner and outer periplasmic TolB? In the full Tot-Pal system, this distinction is abandoned, right? So why introduce it in the first place? See point 6 below.

4. Before “Exchange between differently diffusing states…” you write “this is not mutually exclusive with having most of TolB in the outer compartment in dividing cells (in which transport is localised), which requires D_out to be sufficiently small” I do not understand what “this” refers to and what requires D_out to be sufficiently small. Could you pease clarify?

5. TolA deletion strain. You state that the mobility of Pal remains small in the TolA deletion strain, because Pal is much more abundant than TolB. I find this an interesting finding as it clearly shows the difference between population measurements and the dynamics of individual molecules. To nail this point down, I would love to see an experiment with TolB over expressed in a TolA deletion strain. In that case you should recover the Pal mobility from dividing cells, right?

6. “the increased D_eff of dividing cells is consistent with the model prediction that the mobility of TolB-Pal complexes is greater than that of free TolB.” This goes back to the point about the two diffusion constants D_in and D_out for TolB. Since you have no information about the distance of TolB to the inner and outer membrane, I would prefer you not to talk about inner and outer diffusion constants. Also to talk about free TolB (= not bound to Pal), when it is really bound to TolA was confusing for me. I would suggest that you revise the naming of the different species to help the reader.

7. The authors should provide details of how they numerically solve their equations.

**Have the authors made all data and (if applicable) computational code underlying the findings in their manuscript fully available?**

Reviewer #1: Yes

Reviewer #2: **No: **

Reviewer #3: **No: **The authors do not provide the code for solving the dynamic equations.

PLOS authors have the option to publish the peer review history of their article (what does this mean?). If published, this will include your full peer review and any attached files.

Reviewer #1: No

Reviewer #2: No

Reviewer #3: No
---

## [Decision Letter · Decision Letter 1]

8 Dec 2021

Dear Dr. Murray,

Thank you very much for submitting your manuscript "The quantitative basis for the redistribution of immobile bacterial lipoproteins to division septa" for consideration at PLOS Computational Biology.

As with all papers reviewed by the journal, your manuscript was reviewed by members of the editorial board and by several independent reviewers. In light of the reviews (below this email), we would like to invite the resubmission of a revised version that takes into account the reviewers' comments. In particular, a more detailed comparison between the biophysical basis of postualated model vs diffuse-and-capture models as well and possible mechanistic basis of the role of TolQRA in the model based on the questions raised by the Reviewer 1.

We cannot make any decision about publication until we have seen the revised manuscript and your response to the reviewers' comments. Your revised manuscript is also likely to be sent to reviewers for further evaluation.

Sincerely,

Oleg A Igoshin

Associate Editor

PLOS Computational Biology

Mark Alber

Deputy Editor

PLOS Computational Biology

Reviewer's Responses to Questions

**Comments to the Authors:**

Reviewer #1: The TolQRA localizes at the septum and mobilizes Pal, allowing the dissociated Pal to bind to PG at the septum. As the authors claim, the difference of this model from the "diffusion-and-capture" mechanism is the mobilization of Pal, which allows the "diffusion-and-capture" to take place. What the authors have done is simply assigning the role of TolQRA in the model without any direct mechanistic underpinning. It is ok to make the assumption for modeling. But given that the key notion of the model has already been published in their previous work, I do not see the novelty here on a conceptual level.

The real questions are: How does the TOLQRA mobilize Pal on a mechanistical level? How and why does TolQRA localize to the septum? Are the TolQRA localization and Pal mobilization coupled? I believe these are the real mystery. Mechanistically, the authors did not address any of my key points, which I do believe will level up to make their work suitable for PLoS Comp Biol. Thus, I do not believe that the current manuscript presents a large enough leap in modeling for PLoS Comp Biol.

Reviewer #2: The authors have addressed my concerns and revised the manuscript accordingly.

Reviewer #3: I thank the authors for their answers to the points the other reviewers and I have raised. This has clarified a few points in my mind.

Given the authors’ responses, I still think that it is somewhat misleading to use the term ‘active’ in the abstract. For the equations it doesn’t make any difference, whether this process is active or not (active referring to tearing TolB-Pal complexes apart driven by the PMF). Even though one might sympathise with the authors though that this is the most likely mechanism, it is to my understanding irrelevant to their results. Since all three reviewers commented on this point, I would strongly advise the authors to remove the term ‘active’ from the abstract as not to let this (however well-founded it may be) speculation distract the reader from the solid results of the work.

Reviewer #1 states that the system operates by a typical diffusion-and-capture mechanism. The authors, naturally, have a different opinion. In my opinion, there are clear similarities and clear differences. More importantly to me, the proposed mechanism nicely agrees with the experimental data and thus makes an important contribution to our quantitative understanding of bacterial lipoproteins to division septa. Even if this mechanism is not completely new, I think that it merits publication in PLoS Comp Biol.

**Have the authors made all data and (if applicable) computational code underlying the findings in their manuscript fully available?**

Reviewer #1: Yes

Reviewer #2: Yes

Reviewer #3: Yes

PLOS authors have the option to publish the peer review history of their article (what does this mean?). If published, this will include your full peer review and any attached files.

Reviewer #1: No

Reviewer #2: No

Reviewer #3: No
---

## [Editor Report · Decision Letter 2]

14 Dec 2021

Dear Dr. Murray,

We are pleased to inform you that your manuscript 'The quantitative basis for the redistribution of immobile bacterial lipoproteins to division septa' has been provisionally accepted for publication in PLOS Computational Biology.

Best regards,

Oleg A Igoshin

Associate Editor

PLOS Computational Biology

Mark Alber

Deputy Editor

PLOS Computational Biology

---

## [Editor Report · Acceptance letter]

23 Dec 2021

PCOMPBIOL-D-21-01655R2 

The quantitative basis for the redistribution of immobile bacterial lipoproteins to division septa

Dear Dr Murray,

I am pleased to inform you that your manuscript has been formally accepted for publication in PLOS Computational Biology. Your manuscript is now with our production department and you will be notified of the publication date in due course.

With kind regards,

Katalin Szabo
